# Engineering a Microphysiological Model for Regenerative Endodontic Studies

**DOI:** 10.3390/biology13040221

**Published:** 2024-03-28

**Authors:** Diana Sanz-Serrano, Montse Mercade, Francesc Ventura, Cristina Sánchez-de-Diego

**Affiliations:** 1Department of Dentistry, Universitat de Barcelona, 08907 L’Hospitalet de Llobregat, Spain; diana.sanz@ub.edu (D.S.-S.); montsemercade@ub.edu (M.M.); 2The Bellvitge Biomedical Research Institute (IDIBELL), 08908 L’Hospitalet de Llobregat, Spain; fventura@ub.edu; 3Departament de Ciències Fisiològiques, Universitat de Barcelona, The Bellvitge Biomedical Research Institute (IDIBELL), 08907 L’Hospitalet de Llobregat, Spain; 4Department of Biomedical Engineering, University of Wisconsin–Madison, 550 Engineering Dr, Madison, WI 53706, USA; 5Department of Pathology & Laboratory Medicine, University of Wisconsin–Madison, 1111 Highland Avenue, Madison, WI 53705, USA; 6Carbone Cancer Center, University of Wisconsin–Madison, 1111 Highland Avenue, Madison, WI 53705, USA

**Keywords:** stem cells, apical papilla, microfluidics, cytotoxicity, three-dimensional culture, endodontic irrigants

## Abstract

**Simple Summary:**

Dental pulp infections are common oral health problems that require thorough treatment to disinfect and prepare the root canal using irrigating solutions. However, research on regenerative procedures in endodontics, particularly those involving immature root canals, has been hindered by the lack of suitable laboratory models. In response, this study aimed to create a 3D microphysiological system (MPS) to mimic immature root canals and test the effects of different irrigating solutions. By using human stem cell-derived DSCS cells, researchers found that some irrigating solutions reduced cell viability and affected cell adhesion in the MPS. Notably, this study identified two irrigating solutions, 17% EDTA and 9% HEBP, that showed promising results in terms of cell viability and adherence in the 3D MPS model. These findings emphasize the importance of the MPS for studying root canal treatments and suggest potential alternatives to traditional irrigating solutions for clinical use. This research could lead to improved treatments for dental pulp infections, benefiting patients and dental practitioners alike.

**Abstract:**

Dental pulp infections are common buccal diseases. When this happens, endodontic treatments are needed to disinfect and prepare the root canal for subsequent procedures. However, the lack of suitable in vitro models representing the anatomy of an immature root canal hinders research on regenerative events crucial in endodontics, such as regenerative procedures. This study aimed to develop a 3D microphysiological system (MPS) to mimic an immature root canal and assess the cytotoxicity of various irrigating solutions on stem cells. Utilizing the Dental Stem Cells SV40 (DSCS) cell line derived from human apical papilla stem cells, we analyzed the effects of different irrigants, including etidronic acid. The results indicated that irrigating solutions diminished cell viability in 2D cultures and influenced cell adhesion within the microphysiological device. Notably, in our 3D studies in the MPS, 17% EDTA and 9% 1-hydroxyethylidene-1, 1-bisphosphonate (HEBP) irrigating solutions demonstrated superior outcomes in terms of DSCS viability and adherence compared to the control. This study highlights the utility of the developed MPS for translational studies in root canal treatments and suggests comparable efficacy between 9% HEBP and 17% EDTA irrigating solutions, offering potential alternatives for clinical applications.

## 1. Introduction

Microorganisms and their by-products are the foremost causes of dental diseases, including dental pulp infections [1]. When this happens, endodontic treatments disinfect and recondition the root canal for subsequent filling or revascularization procedures. The final purpose of this process is to maintain the tooth structure intact in order to preserve optimal function. In immature permanent teeth, dental pulp treatment also aims to support continuous root development and apical closure. This process is known as regenerative endodontics [2]. During pulp treatment, irrigating solutions dissolve the necrotic pulp tissue remnants, inactivate endotoxin, and prevent smear layer formation [1]. Recent studies have shown that primary root canal infections are polymicrobial (10–30 bacterial species) in nature and are dominated by obligate anaerobic bacteria [3]; thus, irrigating solutions must have a broad antimicrobial spectrum and be especially effective against microorganisms and biofilms [4]. Sodium hypochlorite (NaOCl) is the standard irrigating solution. NaOCl is used alone or in combination with chelating substances (e.g., ethylenediaminetetraacetic acid (EDTA), citric acid) to chemically clean the inorganic fraction of the smear layer [5], demineralize the dentin extracellular matrix (ECM), and expose a variety of growth factors for regenerative procedures [6]. Although EDTA and citric acid hold the aforementioned favorable properties, they drastically decrease the free chlorine released by NaOCl [7], affecting its antimicrobial and tissue-dissolving activity [8]. Therefore, novel chelating agents with similar properties that do not interact with NaOCl are needed.

In this context, 1-hydroxy ethylidene-1, 1-bisphosphonate (HEBP), also known as etidronic acid, is a non-nitrogen-containing bisphosphonate with anti-bone-resorptive activity [9,10]. Nine percent HEBP eliminates the smear layer to a similar extent as conventional 17% EDTA treatment [11] without eroding the dentin surface [5]. More importantly, HEBP does not affect NaOCl chemical activity [12]. For regenerative endodontics, irrigating solutions must create a suitable niche for stem cell migration and survival. However, the effect of the different irrigating solutions on human apical papilla stem cells (SCAPs) remains unexplored [7,12].

Conventional in vitro dental models predominantly employ two-dimensional (2D) cultures, where cells are grown in monolayers on non-physiological plastic substrates. This approach fails to recapitulate biochemical and physical cues in vivo and the interactions between the cells and the ECM proteins [13,14]. The spatial organization of cells cultured in three-dimensional (3D) structures affects cellular behavior such as proliferation and migration [15]. In the field of endodontics, different devices have been developed for conducting three-dimensional cultures, including the use of dentin discs, organoid-based reconstruction strategies, diffusion chambers, and other animal models, but, independent of the system, there are difficulties in the translation to human models [16]. On the other hand, in vivo experiments can be cost-prohibitive, have low throughput, and require highly specialized personnel [17].

Recently, microfluidic systems have been adapted for 3D cultures. Due to its structure at the micrometer scale, microfluidics offers advantages over traditional methods, such as controlled laminal flow, spatial separation of different cell types, and control of geometry and cell microenvironment [18,19].

Despite the importance of regenerative procedures in endodontics, no 3D cell culture model represents the anatomy of an immature root canal. The aim of this study was to develop a 3D microphysiological system (MPS) based on a luminal structure of 1 mm lined with stem cells from the apical papilla to faithfully reproduce an immature root canal’s structure and to use the model to evaluate the cytotoxicity of new irrigants in a 3D cell culture in vitro.

We hypothesize that microfluidic 3D cultures offer a more accurate representation of real physiology and thus serve as a superior model for preclinical trials. Furthermore, we hypothesize that HEPB could serve as a promising alternative to EDTA, exhibiting lower cell toxicity.

## 2. Materials and Methods

### 2.1. Experimental Conditions

In this study, we evaluated the irrigating solutions shown in Table 1.

All irrigating solutions were freshly prepared in DMEM media, and pH was adjusted to 7 using NaOH or HCl. This adjustment aimed to avoid cytotoxicity resulting from excessively low or high pH levels without affecting the effectiveness of the irrigating solution [20,21]. For the 9% HEBP group, Dual Rinse^®^ (Medcem, Laudongasse, Vienna, Austria) was used and prepared according to the manufacturer’s instructions.

### 2.2. Cell Culture

The Dental Stem Cells SV40 (DSCS) cell line (DSCS) cells were used at passage 25. DSCS is an immortalized cell line from human stem cells from the apical papilla that has previously been characterized [22]. DSCS cells were cultured in DMEM (Sigma-Aldrich, Darmstadt, Germany) supplemented with 10% FBS, 2 mM L-glutamine, and 100 U/mL penicillin/streptomycin. Cells were incubated at 37 °C with 5% CO_2_.

### 2.3. 2D Cytotoxicity Assay

We used 3-(4,5-dimethylthiazol-2-yl)-2,5-diphenyltetrazolium bromide (MTT) to analyze cytotoxicity in 2D. Cells were seeded 1 × 10^5^ DSCS cells/mL in a 12-well plate. After 24 h, DSCS cells were incubated with 500 μL of irrigating solution for 5 min. Then, cells were washed with phosphate-buffered saline (PBS), and 500 μg/mL 3-(4,5-dimethylthiazol-2-yl)-2,5-diphenyltetrazolium bromide (MTT) and 100 μM sodium succinate were added to the culture medium and incubated for 1.5 h at 37 °C for the reaction. Then, the MTT solution was removed, and the solubilization solution (0.5% acetic acid and 10% sodium dodecyl sulfate in dimethyl sulfoxide (DMSO)) was added. Absorbance was read at 570 nm in a microplate reader (Tecan Sunrise Microplate Reader, Tecan, Männedorf, Switzerland). Cell viability was calculated as a percentage of the absorbance for the control group.

### 2.4. Microphysiological Device Design

To overcome the limitation of 2D models, we leveraged the use of PDMS microfluidic devices to reproduce the 3D structure of an immature root canal. Our device utilized the LumeNEXT platform, where a microfluidic chamber was created by two overlapping layers of PDMS. To replicate the immature root canal, we modified the original device dimensions. Specifically, we enlarged the central chamber (∅ 8 mm) to accommodate a 1 mm diameter wax rod. This wax rod served as a mold for generating a luminal structure that accurately reproduced an immature root canal (Figure 1A). Two lateral ports (∅ 3 mm each) flanked the main chamber for collagen injection, while two differently sized ports (∅ 4 mm and 6 ∅ mm) held the wax rod (∅ 1 mm) and were designed to enable passive pumping for fluid flow through the lumen (Figure 1A–C). The device was imaged using a Nikon TI^®^ Eclipse inverted microscope (Nikon Instruments Inc., Melville, NY, USA).

### 2.5. Microphysiological Device Fabrication

Microphysiological devices were fabricated in polydimethylsiloxane (PDMS) using the soft lithography technique for LumeNEXT devices as previously described [19]. Briefly, PDMS was poured over the SU-8 (Microchem Ansell, Richmond, Australia) silicon master molds and then baked at 80 °C for 4 h. After baking, we sandwiched a wax rod of 1 mm diameter in between, serving as a mold for creating a luminal structure that recreated an immature root canal.

Microdevice setup and collagen injection were performed following the previously described protocol [23]. Type I rat tail collagen (Millipore, St. Louis, MA, USA) was prepared in ice at 5.2 mg/mL, pH 7.4. The mixture was injected into the MPS and polymerized for 20 min at 37 °C. Then, the wax rod was pulled out of the device, resulting in a tubular lumen of 1 mm diameter in the collagen gel.

### 2.6. Microphysiological System Biocompatibility

To assess whether the microphysiological device was biocompatible and would support both cell viability and adhesion to the ECM matrix, we lined the lumen with 5 μL of DSCS cell suspension at 2 × 10^7^ cells/mL. To assess cell viability after 48 h, 1:200 calcein staining was performed. Images of four microscopic fields were captured using a fluorescence microscope (Leica DM-IRB, Wetzlar, Germany) and a confocal microscope (Zeiss LSM880, Zeiss, Oberkochen, Germany). The total number of cell nuclei and live and dead cells was quantified using ImageJ.

### 2.7. Small Molecule Diffusion

Rhodamine B (Millipore-Sigma, Missouri, St. Louis, MO, USA) was dissolved at 5 mg/mL in distilled water and later diluted 1:100 in PBS. Rhodamine B was perfused through the central canal of the device and was tracked via fluorescent microscopy (Leica DM-IRB, Wetzlar, Germany) for 5 min. ImageJ software version 1.54i was used to generate fluorescence plot profiles of the central chamber.

### 2.8. Microphysiological Device Treatment and Cell Culture

Lumens were treated following the endodontic revascularization treatment protocol [24]. Lumens were treated with 5 μL of irrigating solution or DMEM for controls (Table 1) for 5 min. Then, lumens were rinsed once with 200 μL of growth media. Then, each lumen was loaded with 5 μL of DSCS cell suspension at 2 × 10^7^ cells/mL. After that, the cell-loaded devices were incubated at 37 °C with 5% CO_2_ for 1 h. Then, lumens were rinsed three times to remove any non-adherent cells.

### 2.9. Cellular Attachment Efficiency and Viability

DSCS cells were cultured overnight in the lumens at 37 °C with 5% CO_2_. Microfluidic devices were stained with 1:200 calcein, 1:1000 propidium iodide (PI), and 1:1000 Hoechst 33342 (Invitrogen, Waltham, MA, USA). Stains were incubated for 15 min at 37 °C. Then, the solution was rinsed with complete growth media. For each device, images of four aleatory microscopic fields were captured using a fluorescence microscope (Leica DM-IRB, Wetzlar, Germany) and a confocal microscope (Zeiss LSM880, Zeiss, Germany). The total number of cell nuclei and live and dead cells was quantified using ImageJ. Cell attachment was quantified as the total number of Hoechst-stained nuclei. Cell viability was determined by calculating the percentage of live cells (calcein-positive) relative to the total number of cells, which was quantified using Hoechst-stained nuclei.

### 2.10. Calcein and CellTracker Staining

DSCS cells were trypsinized and were stained with 1:200 CellTracker Green (CMFDA, Thermo Fisher, Madison, WI, USA). Cells were loaded into the lumens and incubated for 1 h at 37 °C. Lumens were rinsed three times to remove any non-adherent cells. Devices were incubated at 37 °C with 5% CO_2_ overnight. Then, devices were fixed with 4% PFA for 1 h at room temperature and rinsed twice with PBS. Cells were permeabilized with 1% Triton X-100 (Sigma-Aldrich) for 20 min and blocked in 3% bovine serum albumin (Sigma-Aldrich) for 2 h. Double staining with 1:5000 DAPI (Invitrogen) and 1:500 AlexaFluor 633 conjugated phalloidin (Thermo Fisher) was performed. Cell morphology was analyzed under a confocal laser scanning microscope (Zeiss LSM880).

### 2.11. Statistical Analysis

Statistical analysis was performed using GraphPad Prism version 10. First, Q–Q plots were drawn to assess the normality of the samples (Appendix A). Then, statistical analysis was performed using one-way ANOVA with Bonferroni correction. Quantitative data are presented as the mean ± standard error of the mean. Differences were considered significant at * *p* < 0.05, ** *p* < 0.01 and *** *p* < 0.001. All experiments were performed in technical triplicates.

## 3. Results

### 3.1. Endodontic Irrigating Solutions Reduce Cell Viability of DSCS Cells in a 2D Culture Model

In order to determine the suitability of the use of etidronic acid for endodontics, we quantified cell viability in 2D cultures. We incubated DSCS cells in 500 µL of the different irrigating solutions for 5 min, and then we determined cell viability using the MTT assay. We compared the cell viability of DSCS cells after exposure to the different irrigating solutions with DSCS cells. The results demonstrate a tendency for reduction in cell viability for all the irrigants assayed with respect to the control. The decrease in viability was not statistically significant in the group treated with 17% EDTA, where the highest viability was observed, (64.3% of cell viability). In the group treated with 9% HEBP, there was a statistically significant decrease in cell viability compared to the control group (42.6% of cell viability). However, no statistical difference was observed between the effects of 9% HEBP and 17% EDTA. The decrease in viability was statistically significant in the 6% sodium hypochlorite, 1.5% sodium hypochlorite, and 10% citric acid groups, which reduced cell viability to 28.4%, 22.1%, and 8.2%, respectively (Figure 2 and Table 2). These results indicated that all irrigating solutions tend to decrease cell viability on dental papilla stem cells in 2D cultures.

### 3.2. Microphysiological System Biocompatibility

The ideal microfluidic device for endodontic treatments should be biocompatible and support both cell viability and adhesion to the ECM matrix. Therefore, we lined the lumen with DSCS cells and characterized their morphology after 48 h of cell culture. After assembly and loading of stem cells into the collagen matrix, we observed that DSCS cell efficiently attached to the collagen ECM (Figure 3A). Moreover, calcein staining showed that DSCS cells retained up to 90% viability over 48 h of cell culture. Morphology assays with phalloidin staining of the cytoskeleton showed the characteristic spindle-shaped morphology of the SCAPs (Figure 3A,B).

### 3.3. Small Molecule Diffusion

Microfluidic devices must ensure liquids are completely confined in the internal structures of the microdevice, preventing any potential leakage. PDMS has been reported to be permeable to small hydrophobic molecules; therefore, we decided to verify the diffusion of small molecules using Rhodamine B solution. Rhodamine B is a hydrophobic compound that naturally fluoresces in red and has a molecular weight in the range of small molecules (e.g., etidronic acid), which makes monitoring with real-time microscopy possible. We observed that perfused Rhodamine B solution (5 ng/mL) in the central canal (∅ 1 mm) penetrated the collagen of the main chamber, leading to an observable fluorescence front (Figure 4). After 5 min, Rhodamine B diffused 2 cm away from the central canal. We did not observe leakage between the different layers forming the microarray. This suggests that small molecules and lipophilic compounds can diffuse into the collagen matrix and would be retained in the device (Figure 4).

Two lateral ports (∅ 3 mm each) flanked the main chamber for collagen injection, while two differently sized ports (∅ 4 mm and 6 ∅ mm) held the wax rod (∅ 1 mm).

### 3.4. Endodontic Irrigants Affect Cell Adhesion and Viability in Three-Dimensional Culture

After validating the microphysiological device, we analyzed the effects of different irrigating solutions on DCSC cell viability and adhesion. We treated the canal of each device with 17% EDTA, 10% Citric Acid, 9% HEBP, 1,5% NaOCl, and 6% NaOCl or culture media as a control for 5 min. Then, we filled the canals with DSCSs and incubated them for 1 h. Cell adhesion and viability over time were assessed by staining the devices with Hoechst 33342, calcein, and PI. To study cell adherence, we calculated the number of DSCS cells present in each lumen of each condition (Figure 5A). Treatment with irrigating solutions consistently led to a decrease in cell attachment, with statistically significant reductions observed only in the groups treated with 6% sodium hypochlorite, 1.5% sodium hypochlorite, and 10% citric acid. In these cases, the number of attached cells decreased from an average of 1139 cells in the control to 16.33 for 6% sodium hypochlorite, 222.7 for 1.5% sodium hypochlorite, and 453.5 for 10% citric acid. In contrast, in lumens treated with 17% EDTA and 9% HEBP, we found 880.7 and 782.7 cells attached, respectively. The highest cell adhesion was observed in the treatments associated with the 17% EDTA group, followed by the 9% HEBP group (Figure 5B and Table 3).

In parallel, we evaluated the cell viability of the adhered cells in each group. Apart from sodium hypochlorite (NaOCl), there were no notable decreases in cell viability compared to the control (Figure 5B and Table 4), and cell viability remained at 75% or above in all cases except with 6% NaOCl (Figure 5B and Table 4). These results further support that 9% HEBP does not affect cell adhesion or cell viability of the DSCS.

## 4. Discussion

Control of intracanal infection is critical in regenerative endodontics. However, preserving the vitality of SCAPs and promoting their migration, adhesion, and further differentiation inside the immature canal are also essential. Irrigation solutions have the potential to influence cell viability and modify their microenvironment, thus altering their adhesion [25]. The current protocol of the American Association of Endodontics for regenerative procedures recommends the use of sodium hypochlorite at low concentrations (between 1.5 and 3%) combined with 17% EDTA as a chelating factor. Chelating agents play a central role in root canal treatment as they dissolve the smear layer, increase dentine permeability, and induce soluble factors release. EDTA is the most commonly used chelating agent, and, additionally, there is evidence of the suitability of EDTA in pulp revascularization treatments [26] as it promotes cell survival and adhesion to the canal walls compared to other irrigants [27,28]. However, EDTA interacts with NaOCl and decreases its effect by reducing the free available chlorine [29]. Therefore, there is a need to find better combinations of chelating agents and irrigating solutions.

In 2005, Zhender et al. reported that HEBP provided the same level of chelation as 17% EDTA or 10% citric acid without decreasing the level of free chlorine during the first hour [7]. Several studies have shown that the antibacterial capacity of sodium hypochlorite did not decrease when mixed with etidronic acid [30,31,32]. Arias-Moliz et al. [12] compared the effect of 2% peracetic acid, 2% chlorhexidine, and 2.5% sodium hypochlorite alone or in combination with 9% HEBP against Enterococcus faecalis and found that sodium hypochlorite alone or associated with HEBP was the most effective irrigating solution. Moreover, 9% HEBP caused more significant dentin erosion after 2 min of use than 17% EDTA and 2% peracetic acid [5,11,33,34].

Additionally, previous studies indicate that HEBP does not impact cell viability in a 2D fibroblast cell culture [32]. Here, we used a dental stem cell line (DSCS) to analyze the effects of different chelating agents and irrigating solutions on the viability of stem cells of the apical papilla, as they are the main source of stem cells during regenerative endodontic treatments. DSCS cells were derived from human apical papilla stem cells. DSCS cells express human mesenchymal surface markers (e.g., CD73, CD90, and CD105) and retain their trilineage differentiation potential even at late passages, making them suitable for regenerative endodontic studies [9,32]. Unlike previous studies in fibroblast, we observed that 9% HEBP significantly reduced the viability of DSCS cells cultured in 2D, but it did not affect the cell viability of 3D cell culture [32,35]. However, when comparing the effects of 17% EDTA treatment, considered the gold-standard chelating solution, similar outcomes were observed to those of 9% HEBP, with mean cell viabilities for both solutions recorded at 42.55% and 64.5%, respectively, and no significant difference found through a Student’s *t*-test.

While 2D in vitro models are essential tools for preliminary studies, they fail to reproduce the cell microenvironment. Moreover, they do not support migration and cell adhesion to a 3D matrix. De Almeida used 2D microfluidic plates to study the effects of SCAPs on sensory trigeminal neurons [36]. However, to our knowledge, no studies of the effects on SCAPs in 3D cell cultures have been conducted to date, although there is a pressing need for better models that overcome the limitations of 2D cell culture. In order to address these challenges, various 3D systems have been devised. One approach involves employing progenitor cells seeded within 3D scaffolds typically crafted from tooth crown slices [16]. However, this method is expensive, has limited scalability, and relies on the availability of human samples. Alternatively, researchers have utilized cell perfusion chambers and other microfluidic devices to reconstruct the dentine–pulp interface, offering cost-effectiveness, although they do not fully capture the intricate 3D geometry of root canals [16].

Based on the LumeNEXT platform, we successfully developed the first microphysiological 3D model of an immature root canal. The model contains a 1-mm canal that imitates the structure observed in vivo. The MPS relies on PDMS, which is a hydrophobic material that can sequester lipophilic molecules and have an impact on the effective concentration of drugs. We proved that rhodamine diffuses and penetrates into the collagen hydrogel during treatment. Therefore, this platform can be leveraged as a powerful screening tool for evaluating the efficacy of small molecules, both hydrophilic and hydrophobic, in root canal treatment.

Irrigating solutions similarly affect 2D cell viability and 3D cell adherence. The 9% HEBP and 17% EDTA did not significantly affect the capacity of the cells to attach to the collagen hydrogel, suggesting that HEBP could be an alternative to EDTA during regenerative procedures. Interestingly, irrigants had a dramatic effect on the capacity of cells to adhere in the matrix; however, they did not impact cell viability after cells had attached. Irrigating solutions that had a higher effect on cell viability in 2D (10% citric acid and 1.5% or 6% sodium hypochlorite) also reduced the number of cells attached even when they were not directly in contact. Specifically, they reduced cell attachment by 60.18%, 80.45%, and 98.57%, respectively, compared to the control group. These results reinforce the importance of the indirect effect of irrigating solutions on the root canal microenvironment, which will affect cell adhesion [27]. Our findings suggest that, although all chelating and irrigating solutions affect SCAPs cell viability and adherence, both 17% EDTA and 9% HEBP have similar effects. Although additional studies are still required, HEBP should be considered for regenerative procedures. However, it should be noted that all the experiments that were carried out in short-term and longer-time experiments should be performed to study SCAP cell recovery exposure to irrigating solutions. Due to the fact that cells in in vitro experiments (especially in microenvironments or small volumes as microfluidic devices) lack the pH compensation mechanisms present in tissues, all the experiments were performed at pH 7 to avoid cytotoxicity resulting from high or low pH; all substances were adjusted to pH 7. There is evidence that the effectiveness of EDTA- and NaOCl-buffered forms do not lose their effectiveness [20,21]. However, further research should investigate their efficacy at different pH levels to fully understand their potential applications and effects in various environments.

In conclusion, our results confirm that the initial hypothesis demonstrated that the 3D microfluidic device serves as a better model for preclinical trials and that HEPB could serve as a promising alternative to EDTA, exhibiting lower cell toxicity.

## 5. Conclusions

We have developed the first microphysiological 3D model that simulates an immature root canal for in vitro studies. Additionally, our results highlight its potential to analyze the diverse effects of irrigating solutions on apical papilla stem cells. This model represents an interesting first step, as these devices can be modified further and include bioactive elements to evaluate cell behavior, as well as customized to represent more intricate root canals or even study the relationship between different cell types in an environment more akin to that found in vivo.

## Figures and Tables

**Figure 1 biology-13-00221-f001:**
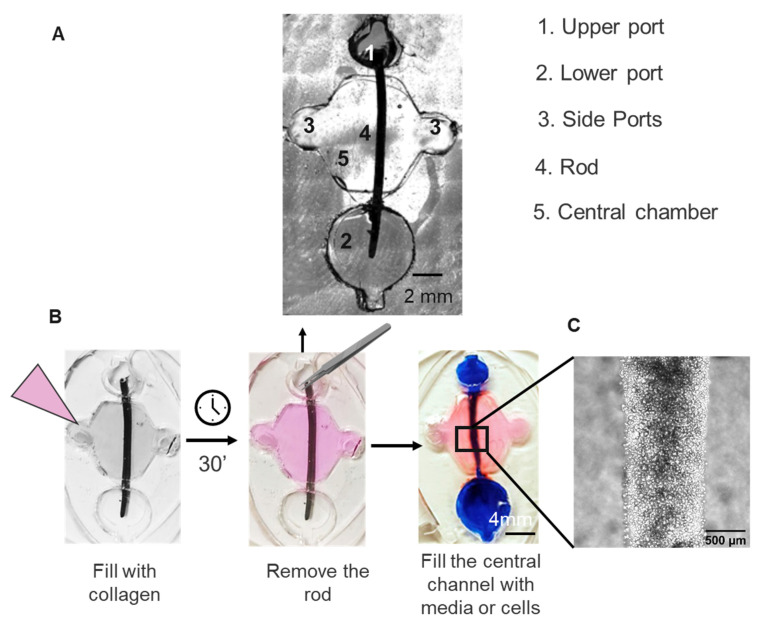
**Microphysiological device for the study of root canal treatments.** (**A**) Microscopy image of the device. (**B**) Schematic representation of the device assembly. The LumeNEXT method was used to create the two-layered microphysiological devices. First, the main chamber was filled with a collagen hydrogel. After collagen polymerization, the rod was removed, generating a canal structure. The canal was filled with a solution containing DSCS cells. (**C**) Image showing DSCS cells filling the root canal.

**Figure 2 biology-13-00221-f002:**
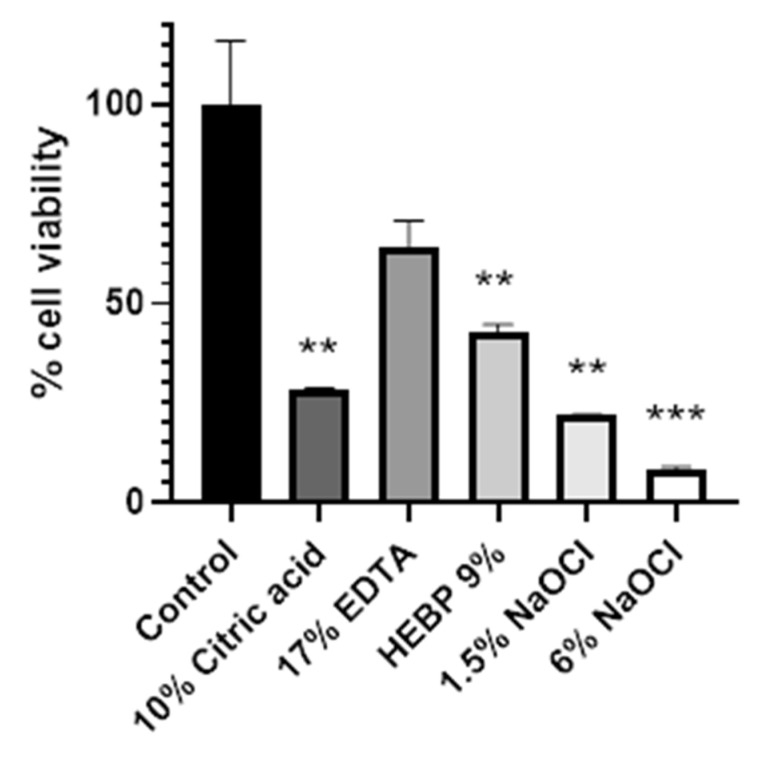
**Effect of irrigating solutions on cell viability in a 2D culture**. Cell viability was calculated as a percentage of live cells with respect to the control in each group. * *p* < 0.05, ** *p* < 0.01, and *** *p* < 0.001 using one-way ANOVA.

**Figure 3 biology-13-00221-f003:**
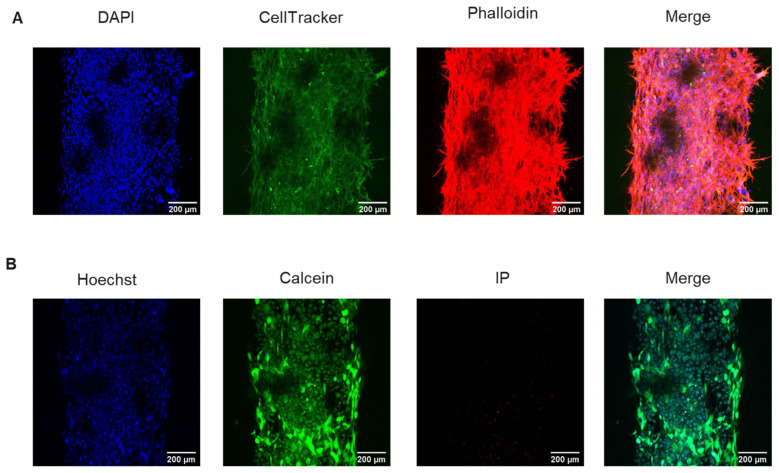
**DSCS cells retain their morphology and viability after 48 h of cell culture in the device**. (**A**) Representative images of DSCS morphology. Cells were stained with DAPI (blue, nuclei), CellTracker Green (green, cell membrane), phalloidin (red, actin cytoskeleton). (**B**) Representative images of cell viability. Cells were stained with Hoechst (blue, nuclei), calcein (green, viable cells), and propidium iodide (red, dead cells).

**Figure 4 biology-13-00221-f004:**
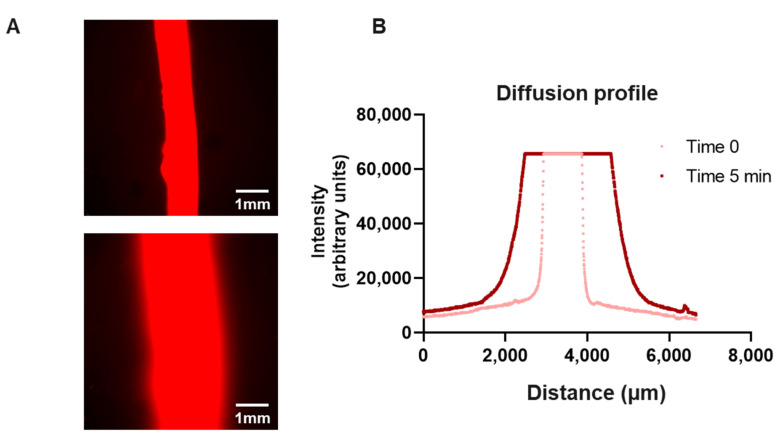
**Diffusion profile**. (**A**) Representative images and (**B**) profile of Rhodamine B diffusion through the device at time 0 and 5 min.

**Figure 5 biology-13-00221-f005:**
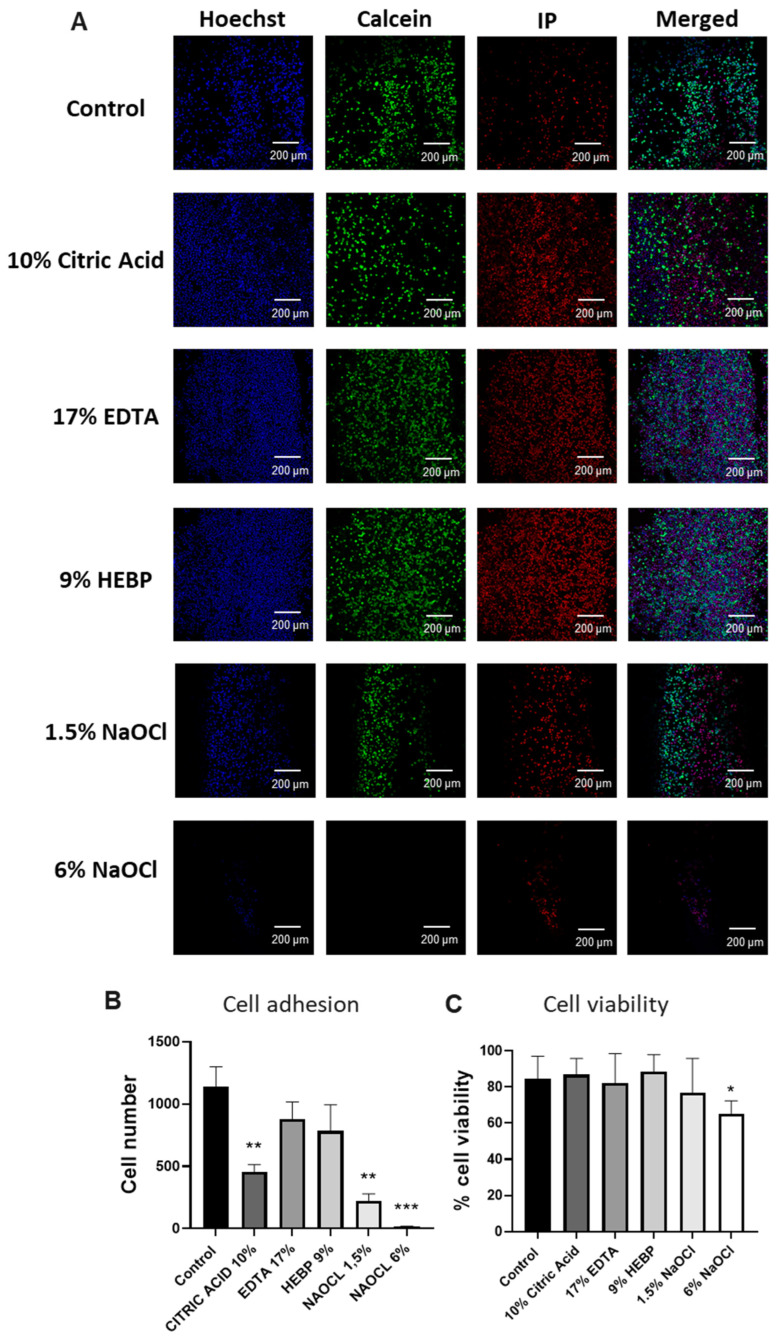
**Effect of irrigating solutions on DSCS cell attachment and viability**. Canals were treated with irrigating solutions for 5 min. Then, canals were washed and filled with DSCS cells. Devices were imaged after 1 h in a confocal microscope. (**A**) Representative images of DSCS cell viability. Green (calcein) = live cells, red (IP) = dead cells, blue (Hoescht) = total nuclei. (**B**) Number of cells adhered in each condition. (**C**) Cell viability of attached cells calculated as percentage of live (calcein) and dead (PI) cells with respect to the total (Hoechst) in each group. * *p* < 0.05, ** *p* < 0.01, and *** *p* < 0.001 using one-way ANOVA.

**Table 1 biology-13-00221-t001:** Irrigating solutions.

Control	DMEM (Merck, Darmstadt, Germany)
Irrigating solutions	17% EDTA (Sigma-Aldrich, Darmstadt, Germany)
	10% Citric acid (Sigma-Aldrich, Darmstadt, Germany)
	9% HEBP (Dual Rinse, Medcem, Laudongasse, Vienna)
	1.5% NaOCl (Panreac, Barcelona, Spain)
	6% NaOCl (Panreac, Barcelona, Spain)

**Table 2 biology-13-00221-t002:** **Mean and standard deviation for cell viability in a 2D culture after treatment with different irrigating solutions**. Cell viability was calculated as a percentage of live cells with respect to the control in each group. The *p*-value was calculated using a one-way ANOVA with respect to the control group.

	Control	10% Citric Acid	17% EDTA	HEBP 9%	1.5% NaOCl	6% NaOCl
Mean	100	28.4	64.32	42.55	22.01	8.22
Std. Deviation	22.57	0.6639	9.129	3.071	0.249	1.17
*p*-value	N/A	0.0019	0.0600	0.0061	0.0012	0.0005

**Table 3 biology-13-00221-t003:** **Effect of irrigating solutions on DSCS cell adherence.** Canals were treated with irrigating solutions for 5 min. Then, the lumens were washed and filled with DSCS cells. Devices were imaged after 1 h in a confocal microscope. Mean and standard deviation for the number of cells adhered to the root canal. The *p*-value was calculated using a one-way ANOVA with respect to the control group.

	Control	Citric Acid 10%	EDTA 17%	HEBP 9%	NaOCl 1.5%	NaOCl 6%
Mean	1139	453.5	880.7	782.7	222.7	16.33
Std. Deviation	450.9	145.8	335.1	515.6	97.43	5.508
*p*-value	-	0.0081	0.6030	0.2979	0.0045	0.0005

**Table 4 biology-13-00221-t004:** **Effect of irrigating solutions on DSCS cell viability.** Canals were treated with irrigating solutions for 5 min. Then, the lumens were washed and filled with DSCS cells. Devices were imaged after 1 h in a confocal microscope. Mean and standard deviation for cell viability calculated as percentage of live (calcein) and dead (PI) cells with control. The *p*-value was calculated using one-way ANOVA with respect to the control group.

	Control	Citric Acid 10%	EDTA 17%	HEBP 9%	NaOCl 1.5%	NaOCl 6%
Mean	87.14	86.82	82.13	88.50	76.63	65.30
Std. Deviation	11.01	8.94	16.37	9.50	19.20	71.01
*p*-value	-	>0.9999	0.9311	0.9998	0.6437	0.0472

## Data Availability

The data presented in this study are openly available in FigShare at DOI https://doi.org/10.6084/m9.figshare.25385482.v1 (accessed on 25 March 2024), https://doi.org/10.6084/m9.figshare.25384987.v1 (accessed on 25 March 2024), and https://doi.org/10.6084/m9.figshare.25383082.v1 (accessed on 25 March 2024).

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
