# Peer review of "Engineering a Microphysiological Model for Regenerative Endodontic Studies"

_biology, 2024, doi:10.3390/biology13040221_

Round 1

Reviewer 1 Report

Comments and Suggestions for Authors

The authors are commended for the efforts put into developing the current manuscript. However, several questions and concerns need addressing to enhance the quality and clarity of the manuscript.

1.      Acronym "DSCS" is used in the text without its full name mentioned anywhere. 

2.     What are the hypotheses tested based on the aim of the study? 

3.     In the methodology section, there is a mention of irrigating solutions being freshly prepared at a pH of 7. Could the authors elaborate on the method of preparation and the rationale behind maintaining this pH level. Additionally, clarification is needed regarding the efficacy of all irrigating solutions used at this pH.

4.     Lines 84 and 146 contain repetitive sentences. Furthermore, there is inconsistency in section numbering, as evident with the absence of numbering for section 2.3.1. Clarification is also required regarding the significance of "D" in section 2.3.2 and its numbering.

5.     The results section lacks numerical values or tables, providing only qualitative descriptions of outcomes in relation to the figures is not adequate. This raises questions about the reliability of the results. It is important to report the values employed to analyse the statistics.

6.     Sections 3.2 and 3.3 contain descriptions more fitting for the methodology section than the results. Only outcomes directly related to the study should be reported in the results section.

7.     The discussion of results is weak and limited without reported values. To enhance coherence, restructuring of paragraphs is also recommended to achieve better flow and cohesion.

Author Response

The authors are commended for the efforts put into developing the current manuscript. However, several questions and concerns need addressing to enhance the quality and clarity of the manuscript.

  1. Acronym "DSCS" is used in the text without its full name mentioned anywhere. 

 We appreciate the comment of the reviewer. DSCS is a cell line developed in the lab. It’s the acronym for Dental Stem Cells SV40. We have added the full name in the summary and line 85 of the manuscript.

  1. What are the hypotheses tested based on the aim of the study? 

We appreciate the suggestion of the reviewer. In order to clarify the aim of the study we have added a sentence describing our main hypotheses in the introduction.

‘We hypothesize that microfluidic 3D cultures offer a more accurate representation of real physiology and thus serve as a superior model for preclinical trials. Furthermore, we hypothesize that HEPB could serve as a promising alternative to EDTA, exhibiting lower cell toxicity.”

  1. In the methodology section, there is a mention of irrigating solutions being freshly prepared at a pH of 7. Could the authors elaborate on the method of preparation and the rationale behind maintaining this pH level. Additionally, clarification is needed regarding the efficacy of all irrigating solutions used at this pH.

We appreciate the reviewer’s comment. We have added the method for preparation of the irrigants in the method section and in the limitation section. In vitro experiments (especially in microenvironments or small volumes as microfluidic devices) require pH buffering mechanisms  to prevent pH alterations  affecting cytotoxicity results. Therefore, all substances were adjusted to pH 7, as there is evidence that buffered forms do not lose their effectiveness.

Granum, P.E.; Magnussen, J. The Effect of PH on Hypochlorite as Disinfectant. Int J Food Microbiol 1987, 4, 183–186, doi:10.1016/0168-1605(87)90025-0.

Serper, A.; Çalt, S. The Demineralizing Effects of EDTA at Different Concentrations and PH. J Endod 2002, 28, 501–502, doi:10.1097/00004770-200207000-00002.

  1. Lines 84 and 146 contain repetitive sentences. Furthermore, there is inconsistency in section numbering, as evident with the absence of numbering for section 2.3.1. Clarification is also required regarding the significance of "D" in section 2.3.2 and its numbering.

 We appreciate the suggestion. We have removed the repeated sentence in line 146. We have fixed the numbering in section 3.2.3. and modify the title to ‘2.3. 2D cytotoxicity assay”

  1. The results section lacks numerical values or tables, providing only qualitative descriptions of outcomes in relation to the figures is not adequate. This raises questions about the reliability of the results. It is important to report the values employed to analyse the statistics.

 We appreciate the reviewer comments. We have added tables for all the graphs in the supplementary information. Additionally, we  comment now the values on the main text. Finally, the data presented in this study is openly available in FigShare at DOI  https://doi.org/10.6084/m9.figshare.25385482.v1, https://doi.org/10.6084/m9.figshare.25384987.v1 and  https://doi.org/10.6084/m9.figshare.25383082.v1

  1. Sections 3.2 and 3.3 contain descriptions more fitting for the methodology section than the results. Only outcomes directly related to the study should be reported in the results section.

 We appreciate the reviewer’s suggestion. We have moved the design of the device to the methodology section. Additionally, we have the sentences that describe the experiment set up on section 3.3.

  1. 7.     The discussion of results is weak and limited without reported values. To enhance coherence, restructuring of paragraphs is also recommended to achieve better flow and cohesion.

We thank the reviewer for the comment. We have restructured the paragraph to enhance coherence. Additionally, we have reported numerical values to improve the discussion. 

Reviewer 2 Report

Comments and Suggestions for Authors

The authors developed a 3D model that mimics an immature root canal (3D microphysiological system, MPS) and used it to study the influence of irrigants on dental stem cell line (DSCS) derived from human apical papilla stem cells (SCAPs) viability and adherence.

The aim of the research is clear and well-formulated.

The obtained results are very interesting as they illustrate cell behavior in spatially organized cell cultures. The data might be used for further regenerative endodontic studies.

Authors also recommended to conduct long-term experiments which will provide additional data and would help to understand deeply the effect of irrigating solutions on SCAP cells.

The following comments do not diminish the value of the Article:

 Line 11 Probably it would be better to decipher the abbreviature of the cell line - DSCS.

Line 15 It would be better to decipher the abbreviature – HEBP.

Line 44 Probably it would be better to place the abbreviature closer to the full title of the substance.

Line 89 Is ‘D cytotoxicity assay’ for ‘2D cytotoxicity assay’?

Line 129 It would be better to indicate the period of cell incubation inside the device.

Line 148 The fragment ‘500 l’ should be specified.

Line 149 The abbreviation ‘DCSC’ should be specified.

Line 155 As there were no significant differences probably it would be better to mention the tendency when describing the effect of 17% EDTA or 9% HEBP.

Line 159 It would be good to include the formulation for cell viability calculation in the Materials and Methods section.

Line 177 Probably it would be better to indicate the equipment that was used to obtain microscopy image of the device.

Line 187 It would be better if the method of the cellular attachment efficiency assessment would be described.

Lines 188-189 It would be better to include the explanation of cell viability calculation using calcein staining.

Line 190 Description of the phalloidin staining procedure should be added to the Materials and Methods section.

Line 194 The characters A, B should be indicated in the Figure 3.

Line 195-196 The information about Cell Tracker Green staining used during the research should be added to the Materials and Methods section.

Line 217 The Graph should be checked, distance's measure units should be checked. It would be better also to specify the sizes of the device.

Line 225 It would be better to specify which MSCs were used.

Line 227 Would you please specify how cellular adherence was studied?

Lines 243, 244 Probably it would be better if it would be specified why the specific procedure was chosen: treating the canals with irrigating solutions for 5 min and then washing the canals and filling them with DSCS.

Line 247 It would be better if the procedure of cell viability calculation would be included in the Materials and Methods section.

Please check, the references should be described according to the Journal's requirements.

Comments on the Quality of English Language

Academic English language is used.

Author Response

Reviewer 2

The authors developed a 3D model that mimics an immature root canal (3D microphysiological system, MPS) and used it to study the influence of irrigants on dental stem cell line (DSCS) derived from human apical papilla stem cells (SCAPs) viability and adherence.

The aim of the research is clear and well-formulated.

The obtained results are very interesting as they illustrate cell behavior in spatially organized cell cultures. The data might be used for further regenerative endodontic studies.

Authors also recommended to conduct long-term experiments which will provide additional data and would help to understand deeply the effect of irrigating solutions on SCAP cells.

 The following comments do not diminish the value of the Article:

 Line 11 Probably it would be better to decipher the abbreviature of the cell line - DSCS.

 We appreciate the comment of the reviewer. DSCS is a cell line developed in the lab. It’s the acronym for Dental Stem Cells SV40. We have added the full name in the summary and line 85 of the manuscript.

Line 15 It would be better to decipher the abbreviature – HEBP.

We appreciate the comment. We have included the full name of HEBP in the abstract and line 44.

Line 44 Probably it would be better to place the abbreviature closer to the full title of the substance.

We appreciate the comments, we have placed the abbreviature next to the full name of the substance.

Line 89 Is ‘D cytotoxicity assay’ for ‘2D cytotoxicity assay’?

The reviewer is right. We have modified the title.

Line 129 It would be better to indicate the period of cell incubation inside the device.

We appreciate the suggestion. We have included that information.

Line 148 The fragment ‘500 l’ should be specified.

Thank you for the comment. We have corrected the units.

Line 149 The abbreviation ‘DCSC’ should be specified.

Thank you for the comment, we have corrected the name DCSC to DSCS.

Line 155 As there were no significant differences probably it would be better to mention the tendency when describing the effect of 17% EDTA or 9% HEBP.

We appreciate the suggestion. We have modified that section accordingly to emphasize the tendency.

Line 159 It would be good to include the formulation for cell viability calculation in the Materials and Methods section.

We appreciate the suggestion. We have included that information in line 104.

“Cell viability was calculated as a percentage of the absorbance for the control group”

Line 177 Probably it would be better to indicate the equipment that was used to obtain microscopy image of the device.

We appreciate the comment. We have included that information in line 117.

Line 187 It would be better if the method of the cellular attachment efficiency assessment would be described.

We appreciate the comment. We have included this information in method section line 161

Lines 188-189 It would be better to include the explanation of cell viability calculation using calcein staining.

We thank the reviewer for the comment. We have clarified in line 168 the methodology used to calculate cell viability.

“Cell viability was determined by calculating the percentage of live cells (calcein-positive) relative to the total number of cells, which was quantified using HOECHST-stained nuclei.” 

Line 190 Description of the phalloidin staining procedure should be added to the Materials and Methods section.

We appreciate the comment. We have included a new section in materials and methods describing the procedure (Line 173).

“DSCS were trypsinized and were stained with 1: 200 CellTracker Green (CMFDA). Cells were loaded into the lumens and incubated for 1h at 37°C. Lumens were rinse three times to remove any non-adherent cells. Devices were incubated 37 °C with 5% CO2 overnight. Then, devices were fixed with 4% PFA for 1h at room temperature and rinsed twice with PBS. Cells were permeabilized with 1% Triton X-100 (Sigma–Aldrich) for 20min and blocked in 3% bovine serum albumin (Sigma–Aldrich) for 2h. Double staining with 1: 5000 DAPI (Invitrogen) and 1:500 AlexaFluor 633 conjugated phalloidin (Thermo Fisher) was performed. Cell morphology was analysed under confocal laser scanning microscope (Zeiss LSM880).”

Line 194 The characters A, B should be indicated in the Figure 3.

We appreciate the comment. We have modified the figure accordingly.

Line 195-196 The information about Cell Tracker Green staining used during the research should be added to the Materials and Methods section.

We appreciate the comment. We have included it in materials and methods (line 173)

Line 217 The Graph should be checked, distance's measure units should be checked. It would be better also to specify the sizes of the device.

We appreciate the reviewer comments. Reviewer was right, the units of the distance were wrong. We have corrected the mistake. We have also increased the size of the scale bar in part A. Additionally we have included a sentence specifying the sizes of the device.

Line 225 It would be better to specify which MSCs were used.

We appreciate the reviewer’s comment. We have clarified that the type of MSCs used were the DSCS. Only DSCS cells were used in this study.

Line 227 Would you please specify how cellular adherence was studied?

We appreciate the comment. We have included this information in method section line 161

2.8. Cellular attachment efficiency and viability Fluorescence microscopy in 3D

“DSCS were cultured overnight in the lumens at 37 °C with 5% CO2. Microfluidic devices were stained with 1: 200 calcein, 1: 1000 propidium iodide (PI) and 1:1000 Hoechst 33342 (Invitrogen, Massachusetts, USA).  Stains were incubated for 15 min at 37°C. Then, the solution was rinsed with complete growth media. For each device, images of four aleatory microscopic fields were captured using a fluorescence microscope (Leica DM-IRB, Wetzlar, Germany) and confocal microscope (Zeiss LSM880, Zeiss, Germany). The total number of cell nuclei, live and dead cells was quantified using ImageJ. Cell attachment was quantified as the total number of DAPI-stained nuclei. Cell viability was determined by calculating the percentage of live cells (calcein-positive) relative to the total number of cells, which was quantified using DAPI-stained nuclei.”

Lines 243, 244 Probably it would be better if it would be specified why the specific procedure was chosen: treating the canals with irrigating solutions for 5 min and then washing the canals and filling them with DSCS.

In endodontic revascularization treatment protocols (5,6), irrigants are applied for 5 minutes inside root canal, so that duration of action is what was used in our study to mimic the clinical situation.

  1. ICES. AAE Clinical Considerations for a Regenerative Procedure Revised 5/18/2021. 2021;(March):1–19.
  2. Galler KM, Krastl G, Simon S, Van Gorp G, Meschi N, Vahedi B, et al. European Society of Endodontology position statement: Revitalization procedures. Int Endod J. 2016;49(8):717–23.

Carbon Cancer Center

University of Wisconsin–Madison   1111 Highland Avenue   Madison, Wisconsin 00000

608-982-5268   Email: csanchezdg@gmail.com

We have included this information in the methodology section.

Line 247 It would be better if the procedure of cell viability calculation would be included in the Materials and Methods section.

“Cell viability was determined by calculating the percentage of live cells (calcein-positive) relative to the total number of cells, which was quantified using DAPI-stained nuclei.” 

 We have clarified in line 168 the methodology used to calculate cell viability.

Please check, the references should be described according to the Journal's requirements.

We have updated the references according to the journal requirements.

Your faithfully,

Cristina Sanchez de Diego

Round 2

Reviewer 1 Report

Comments and Suggestions for Authors

There are still repetitive sentences pasted in different paragraphs. Lines 322-323 and 345-37 "irrigating solutions, directly and  indirectly, affect SCAPs by affecting their viability and altering the environment, which could affect cell adhesion.

What is the outcome of the hypotheses listed at the end of the introduction section. This should either be accepted or rejected in accordance with the result as a sentence and should form part of the discussion.

I will recommend the tables be included as an integral part of the manuscript and not placed in a supplementary file.

Also the references are not adequately formatted.

Author Response

There are still repetitive sentences pasted in different paragraphs. Lines 322-323 and 345-37 "irrigating solutions, directly and  indirectly, affect SCAPs by affecting their viability and altering the environment, which could affect cell adhesion.

We appreciate the reviewer comment. We have modified the sentence line 306.

What is the outcome of the hypotheses listed at the end of the introduction section. This should either be accepted or rejected in accordance with the result as a sentence and should form part of the discussion.

We appreciate the suggestion. We have included that sentence at the end of the discussion.

I will recommend the tables be included as an integral part of the manuscript and not placed in a supplementary file.

We appreciate the recommendation. We have include the tables on the main document

Also the references are not adequately formatted.

We appreciate the comment. We have formatted the references